# Strain Monitoring of a Composite Drag Strut in Aircraft Landing Gear by Fiber Bragg Grating Sensors

**DOI:** 10.3390/s19102239

**Published:** 2019-05-15

**Authors:** Agostino Iadicicco, Daniele Natale, Pasquale Di Palma, Francesco Spinaci, Antonio Apicella, Stefania Campopiano

**Affiliations:** 1Department of Engineering, University of Naples “Parthenope”, Centro Direzionale Isola C4, 80143 Napoli, Italy; pasquale.dipalma@uniparthenope.it (P.D.P.); campopiano@uniparthenope.it (S.C.); 2Infratel Italia S.p.A, 00187 Roma, Italy; daniele.natale@uniparthenope.it; 3Magnaghi Aeronautica SpA, via Galileo Ferraris, 76, 80142 Napoli, Italy; fspinaci@magnaghiaeronautica.it (F.S.); aapicella@magnaghiaeronautica.it (A.A.)

**Keywords:** Fiber Bragg gratings, fiber optic sensors, aircraft landing gear, load monitoring system, composite device

## Abstract

This work reports on the use of Fiber Bragg Grating (FBG) sensors integrated with innovative composite items of aircraft landing gear for strain/stress monitoring. Recently, the introduction of innovative structures in aeronautical applications is appealing with two main goals: (i) to decrease the weight and cost of current items; and (ii) to increase the mechanical resistance, if possible. However, the introduction of novel structures in the aeronautical field demands experimentation and certification regarding their mechanical resistance. In this work, we successfully investigate the possibility to use Fiber Bragg Grating sensors for the structural health monitoring of innovative composite items for the landing gear. Several FBG strain sensors have been integrated in different locations of the composite item including region with high bending radius. To optimize the localization of the FBG sensors, load condition was studied by Finite Element Method (FEM) numerical analysis. Several experimental tests have been done in range 0–70 kN by means of a hydraulic press. Obtained results are in very good agreement with the numerical ones and demonstrate the great potentialities of FBG sensor technology to be employed for remote and real-time load measurements on aircraft landing gears and to act as early warning systems.

## 1. Introduction

The aerospace sector is constantly searching for new solutions to optimize the component lifetime and the maintenance of aircraft. Research efforts on smart-martials, intelligent systems and/or innovative monitoring technologies are widely welcome. In this context, the aircraft manufacturers started to use composite materials in aircraft components since the early 1980s [1]. Composite materials are a key solution in aircraft structures due to the lightness of composites and the corrosion problem in aluminum or metallic items. However, the introduction of novel materials requires appropriate systems for monitoring of their health state before enabling a wide use [2].

The landing gear (LG) system is an important component in aircraft [3]. There are several types of LGs and arrangements of LGs in aircraft: the most common LG arrangement is the tricycle-type one. The LG aims to provide a suspension system during the take-off and landing phases and during the taxi operation. It is designed to absorb and dissipate the energy of landing impact. Moreover, the LG facilitates braking of the aircraft and provides directional control of the aircraft on ground using a wheel steering system [1,3]. From these considerations, the LG must be an extremely resistant structure. At same time, it should be as light as possible because it is not used for most of the flight time [3]. On this line of argument, LGs realized in composite material represent a unique solution if the mechanical robustness can be kept. To address this ambitious goal, appropriate structural health monitoring (SHM) systems both for the testing and for real operation phases are welcome.

The current SHM technology used in composite materials is done using sensors, especially optical sensors [4,5,6,7,8]. This is since the traditional strain gauge is sensitive to lightning, current leakage and corrosion, whereby inaccurate readings will be shown. On contrary, in last 20 years the potential of optical sensors has been widely demonstrated [9]. Optic sensors provide several advantages for aerospace applications that include insensitivity to electromagnetic interference, lightweight and flexible harness, multiplexing and multi-parameter sensing, high measurement accuracy, low power requirements per sensor, remote interrogation and operation, and the potential to embed in composite structures. Among the several optical fiber sensor configurations [10], Fiber Bragg Sensors (FBGs) are considered the favorite technology [4,5,6,7,8] for their capability to build a dense multiplexing SHM network, durability under extreme weather conditions and ease of application to the surfaces of different structures. As matter, FBG sensors represent the most mature and assessed sensing platform ready to be used in real industrial scenarios [4,5,6,7,8,11,12].

Several dozens of FBG sensors, including temperature and strain can be written on the same optical fiber, and can be simultaneously interrogated by a single interrogator unit. A very important point in FBG based SHM system is the method to couple the FBG sensors with the composite structure, and different ways to accomplish this have been experimented [10]. The most used and simple ways consist in the FBG surface bonding with the composite structure previously created. There is a big literature knowledge [13] that can help to identify the best glue and cure procedure for the specific application: some authors reported that the most important factors are the adhesive thickness and the length of bonding [14]. A second most integrated way to apply the sensors is to insert the FBG between two composite structures linked together, but this method is possible only in situations that provide for assembly of different parts. The last and more integrated method to apply the FBG to the structure is incorporation of FBG during the composite production, in order to place the FBG just some layers underneath the surface and ensure the sensors network from surface accidental contacts [1] and detect damage in different locations inside the material [15].

In this context, this manuscript deals with the activities carried-out within the framework of a National Projects (PON03PE 00135, “CAPRI - Carrello per Atterraggio con Attuazione Intelligente”) aimed to identify novel composite items of a standard LG and appropriate fiber optic sensors to investigate its SHM during loading testing. In particular, in this paper we present the successful use of several FBG sensors for the real-time, continuous and remote monitoring of the strain profile of novel composite items of an aircraft LG during increasing loads test. We designed and subsequently tested several FBG strain sensors installed on a true composite item of a real LG, provided by the company Magnaghi Aeronautica Spa. To optimize the localization of the FBG sensors, load condition was numerically studied by finite element method (FEM) numerical analysis. Successively, several experimental tests have been done in range 0–70 kN by means of a hydraulic press, up to the breaking of the innovative item. Obtained results are in very good agreement with the numerical ones and demonstrate the great potentialities of FBG sensors technology to be employed for remote and real-time load measurements on aircraft landing gears. It is worth noting that in most of literature works FBG sensors were proposed to sensorize large components such as turbines, wings or hull elements that have a large dimension. Moreover, recent works successfully demonstrate the use of FBG sensors to detect defects or cracks in mechanical structure even with reduced size [16,17,18]. In our work, the FBG sensors are successfully applied on complex geometry characterized by a high surface curvature.

## 2. Materials and Method

### 2.1. Composite Item to Be Sensorized

The design of the landing gear, according to the Airworthiness Regulations, must take into account several requirements in terms of safety, strength, stability, etc., under all possible in-service loading conditions (weather conditions included). The landing gear is one of the main structural component characterizing an aircraft. It is aimed to support the aircraft during the landing, the tacking off and ground operations. Among such loading conditions, the landing phase defines the design specifications, since it is the most burdensome. As a result, this loading condition determines its structural size.

Most of LG components are currently steel made in order to provide a robust suspension system during the take-off and landing phases. The design and fabrication of LG items in composite materials represent open challenges offering significant lightening of the landing gear. Within the Italian Project CAPRI, the project leader Magnaghi Aeronautica Spa proposed and selected the drag brace lower arm of the nose landing gear of the Alenia C27J aircraft (plotted in Figure 1a) as a target sample to be realized in composite materials.

The aim was to realize a composite material drag brace and demonstrate the capability to use optical fiber sensors as a real-time strain state monitoring system. The innovative item was proposed (see Figure 1b) and realized (see Figure 1c) by Hexply 914-40%-G803, or similarly Cytec 977-2A/HTA. The total weight was less than 1.0 Kg when compared with the steel version of about 3.0 Kg. Concerning the fixing points with the rest of the LG, it presented an upper fixing point (A) equipped with metallic ring with inner diameter D_A_ = 52.7 mm. The body length was L = 162.0 mm. The down fixing point consisted of two arms with lengths L_B_ = 87.0 mm and a separated width of W_B_ = 50.0 mm. The arms are equipped with metallic rings, in following named B1 and B2, with inner diameters D_B_ = 32.7 mm. The shape created a bent surface between B1 and B2 with a bending radius of 24.0 mm, which was critical for strain surface distribution.

### 2.2. FBG Sensors

A Fiber Bragg Grating sensor represents a robust and efficient sensor for precise determination of the deformations of the selected item during loading test. A FBG consists in a periodic modulation of the core refractive index along the core of a standard single-mode optical fiber with typical length of 1–20 mm. Consequently, FBG reflects a specific wavelength, called Bragg wavelength λ_B_ that depends on the period Λ of the perturbation according to the following formula [7,11]:
(1)$$\lambda_{B}=2n_{\mathrm{eff}}\Lambda$$
where n_eff_ is the effective refractive index of the guided core mode.

The effective refractive index of the core and the spatial periodicity of the grating are both affected by changes in strain (Δε) and temperature (ΔT). As a result, the Bragg wavelength changes its position, according to the following equation:
(2)$$\frac{{\Delta \lambda }_{B}}{\lambda_{B}}=S_{T}\Delta T+S_{\varepsilon }\varepsilon$$
where S_T_ is the thermal sensitivity coefficient and S_ε_ is the strain sensitivity coefficient. According to Equation (2), strain and temperature may be deduced from the measurement of the Bragg wavelength shift and strain and temperature contributions can be separated through the implementation of specific sensing configurations. For our purposes, FBGs were employed for accurate measurement of the strain profile of the structure where they are embedded. Additionally, the thermal effect has been properly compensated for by using an additional free and unstrained FBG, so that it only recorded temperature changes [7].

Moreover, it is worth noting that in most SHM applications, bare FBGs are too fragile to be used as robust solutions, especially when large stresses or loads are applied. In such cases, FBGs integrated in protective packages, commercially available, are often preferred. However, most packages significantly increase the final size of the sensors and thus are undesired for SHM of small items with complex geometry. In our case, the most stringent requirement was keeping the sensing area as small as possible in order to be able to investigate the strain profile in a small region with fast changes of the strain profile. Thus, the attention was focused on unpackaged FBGs with lengths of 1.0–10.0 mm depending on sensors layout. All sensors were protected by a tight polyamide coating, 20 µm thick. During load test, the Bragg wavelengths of all sensors as function of the item load stress were detected by Bragg meter (from Fiber sensing) with eight channels and operating in range of 1500–1600 nm with a scanning rate of ½ Hz and wavelength resolution of 1.0 pm.

### 2.3. Sensors Bounding Procdure and Characterization

The use of FBG as optoelectronic strain gauge sensors is well known [4,5,6,7,8] However, it demands specialist solutions for each application field. The gluing of sensors with items needing to be sensorized represents the most critical issues to be carefully addressed to favor maximum strain transfer to sensors. This section presents and discusses preliminary experimental tests aimed to identify the more appropriate adhesive to fix the fiber optic sensors to the composite surface. We considered standard electric strain gauges (SGs), provided from Micro-Measurements with well-known gauge factors as references. We also investigated the effects of different adhesives on the response of electric SGs.

The preliminary tests were provided by a sensorized composite panel with dimensions of 40 cm × 20 cm with one end (20 cm long) fixed, as schematically illustrated in Figure 2a. On the top surface of the panel, several electric SGs and several FBGs were fixed at same distance from the wall by different adhesives while incremental loading force (five steps from zero to about 470 N) was applied on the opposite side acting as significant strain on the bounded sensors.

Concerning SGs, different kind of adhesives were preliminarily tested; the best results have been obtained by using cyanoacrylate-based adhesives, according with [19,20]. Figure 2b plots the strain profiles of two SGs fixed by different commercial cyanoacrylate adhesives, Micro-Measurements M-Bond 200 (more elastic feedback), and Loctite Super Attak (more rigid feedback), respectively, when the composite panel was subject to increasing and decreasing step-by-step force sequence. The Loctite Super Attak provided a better strain transfer, recording a higher strain measurement of about 15% as compared with M-Bond glue. Maximum strain surface of almost 2000 µɛ was measured by a load force of 471 N. Similar analysis and conclusions were achieved concerning FBG sensors. Consequently, in the following tests all electrical and optical sensors were fixed by Attack adhesive.

Moreover, Figure 3a,b plot the spectra of a 1.0 mm long FBG and of a 10.0 mm long one, respectively, fixed to the composite panel by means of Attack adhesive in the different load conditions. The maximum applied load inducing a superficial strain of about 2000 µɛ forced Bragg wavelength red shifts of 2.54 nm and 2.42 nm for the short and long FBG, respectively. It is important to highlight that despite the high strain values, the shape and maximum reflectivity of the FBGs were unchanged, confirming that the bonding process was able to transfer a uniform strain state to the sensors.

Finally, the FBGs’ strain sensitivity was estimated by the comparison of electrical and optical sensors. First, the deformation profile of the used panel has been displayed in Figure 4a where the strain values of the electrical sensor were plotted versus the applied force: it exhibited a linear behavior with slope of 4.15 με/N. Instead, Figure 4b plots the relative wavelength shifts of the short and long FBG versus the surface strain: the behavior is absolutely linear over the range of 0–2000 με. Moreover, as expected, the sensitivity values for the 1 mm and the 10 mm long FBGs were practically the same, with negligible differences probably inducted by the experimental measurements and the position alignment between the two FBG of different sizes on the panel surface: the experimental sensitivity was estimated to be $S_{strain}=\left(8.2\pm 0.05\right)\cdot 10^{-7}\;\left[{\mu \varepsilon }^{-1}\right]$.

### 2.4. Identification of the Best Locations for Sensors Installation

This section deals with the identification/selection of the locations to be sensorized on the item surface. This analysis should take into consideration the most stressed regions or the geometric characteristics.

To this aim, Magnaghi Aeronautica provided a static stress numerical analysis of the composite item by means of the FEM. The model permits the estimation of the strain distribution along the main axis of the item structure when subjected to the load. Figure 5 plots the FEM result in terms of strain surface along *y*-axis when the item is loaded with 50 kN. It is worth highlighting that the numerical model does not take into consideration metallic ring in the fixing points and thus it is reasonable to believe that the numerical results close the fixing points can be different from the real ones. However, a more accurate FEM model is far from the aim of this work.

From data reported in Figure 5, we designed a custom FBG array, Y-array, aimed to measure the y-strain profile, as schematically plotted in Figure 6a. Y-array, including six FBG sensors, was arranged on the center of the lateral surface to measure the longitudinal (along *y*-axis) strain profile. The sensors were non-uniformly spatially distributed depending on numerical strain profile slope. Sensors Y_1_, Y_2_, Y_3_, Y_4_, Y_5_ and Y_6_ were thus positioned along *y*-axis at 5 mm, 10 mm, 22 mm, 38 mm, 70 mm and 135 mm, respectively.

Finally, a four FBG sensor array, C-array, was designed by the analysis of geometric characteristics. It took into consideration the bending region and thinning of the arms forming the double fixing point, B1 and B2. C-array was designed to measure the surface strain in the bent region. It was selected to demonstrate the goodness of the bonding procedure and the potentiality of the FBG technology in monitoring of surface strain on composite items with complex geometrical shapes. The sensors of C-array, C_1_, C_2_, C_3_ and C_4_ were equally distributed along the *c*-axis, starting from the center of the bent region and separated by 10 mm (as one can straighten the bent region along the *c*-axis). The sensor C_1_ was in the center of the bent region whereas the C_4_ sensor was the closest to the fixing point. In the Cartesian-system plotted in Figure 6a, the bent *c*-axis moved in the plane xy for z = −19.5 mm.

All sensors were glued by means of the procedure discussed in previous section and consisted of unpackaged FBGs with 20 µm thick polyamide coating. Moreover, most of the sensors were based on 1.0 mm long FBGs, in order to decrease the sensing area. Only the sensors Y5 and Y6 were selected to be 10.0 mm long because they are expected to be fixed in a region with low strain variation, as from numerical results. Figure 6b plots the spectra of all FBG sensors: Y-array and C-array. All sensors 1.0 mm long exhibited a FWHM (Full Width at Half Maximum) bandwidth of about 1.6 nm whereas the 10.0 mm long gratings showed a bandwidth of about 0.2 nm. Figure 6c–e shows some pictures during the gluing of sensors. Moreover, the fiber coming out from the composite item was then protected by a red silicone glue.

Finally, one more FBG sensor, not fixed to the item, was included for the compensation of the thermal changes during the experimentation. To this aim we supposed that the item as well as all the glued FBGs were in the same thermal state. During the temperature test variations lower than 3 °C were measured.

### 2.5. Experiemntal Setup

The real-time structural monitoring of the composite item consisted of a continued monitoring of the FBG arrays response when the item was subjected to an incremental load up to 50 kN. This permitted the emulation of a true stress situation for the counterattack, replicating a typical traction during the landing phase of the vehicle.

The response of the FBG arrays was measured by means of FBG interrogation unit (Bragg meter) exhibiting eight-channel operating in a range 1500–1600 nm and with scanning rate of up to 1/2 Hz. The Bragg meter communicated with a standard notebook by a custom software that permited the acquisition of full-spectra of the sensors or Bragg wavelengths of all sensors.

Finally, the loading test was carried out by means of an appropriate press machine for high load condition provided from a Magnaghi co-worker. The Figure 7a,b shows pictures of the sensorized item fixed in the load-machine by means of two proper metallic holders (upper and lower ones). The load-machine was programmed to apply an incremental traction with a rate of 250 N/s.

## 3. Results and Discussion

This section presents and discusses the experimental results, in terms of responses of FBG strain sensors, when the drag brace item is loaded by a longitudinal force linearly incrementing with rate of 250 N/s. In particular, comparative analysis of the behavior of the Y- and C-array in different sections is take into consideration.

Figure 8a plots the strain values returned from the Y-array and C-array, versus acquisition time during the first test section, named test A. The applied force starts from zero (at 10 s) and reaches the maximum load of 25 kN at 110 s. During the incremental load ramp, as expected, all sensors of Y-array explore positive longitudinal strain/traction. In particular, the measured strain increases moving from Y_1_ to Y_5_, accordingly with numeric results and with the decreasing of the cross-section area of the composite item. On the contrary, the strain measured from Y_6_ is significantly lower than Y_5_ one. It can be attributed to the further increase of the item cross section (as moving from the Y_5_ section to the Y_6_ section) and to the proximity to the metallic ferule of the upper fixing point.

The C-array monitors a much more critical region of the item due to the rapid change of the surface strain state and its geometry shape. The sensor C_1_, fixed in the center of the bent region between the ferules B1 and B2, returned a negative response (compressive strain). Due to its location, it measures strain along *x*-axis, which is compressed when the item is longitudinally stretched. Moving along the bent region, thousands of C_2_, C_3_ and C_4_ sensors, the compressive effect of the *x*-axis decreases and the longitudinal elongation along y direction appears. The sensor C_4_ measures the highest positive strain.

Finally, after 110 s the maximum load of 25 kN was kept unchanged for several minutes (up to 1020 s). During this interval, all sensors showed unchanged response as well. Furtherly, the item was suddenly unloaded and thus all sensors recovered their original unperturbed response. The sensor C_4_ showed atypical additive noise, as compared with other sensors, that we were not able to understand.

The Figure 8b plots the response of all sensors in the second test section, test B. Here the applied force linearly increments up to 50 kN (at about 210 s) with same rate. All sensors measure strain increasing with the applied force and the relative behavior of all sensors is in good agreement with the previous test section. However, at 170 s (approximately 40 kN) an anomalous behavior was observed for the sensors of C-array, especially for C_3_ and C_4_: they explore an improvised reduction of the strain of about 110 με and 420 με, respectively. Furtherly they remain almost unchanged up to the end of the test section. It is worth noting that, there was no apparent reason to understand such behavior (at 170 s) because the composite item does not show visible damages and sensor integration with composite surface were still good. However, it is obvious that the sudden response change acts as an early warning since it can be related to a partial breaking of bulk composite layers of the composite item (not visible on external surface). Note that this apparently unfounded hypothesis has been successively confirmed.

Concerning other sensors, the returned strain values continued to increase (decrease for C_1_ and C_2_ reading a compressive state) up to 210 s when the maximum load of 50 kN was reached. Successively, the input load as well as the measured strain were unchanged for several minutes (up to 1120 s). Finally, all sensors recovered to the original unperturbed value when the load state was removed. Concerning the sensors C_4_ the behavior of the recovery curve is like the previous case.

Finally, Figure 8c shows the returned strain values during C section. It was planned to reach a maximum load of 75 kN, significantly higher than the expected working load. Strain measured from all sensors starts the linear shift in perfect agreement with previous experiments. However, after 282 s of the incremental load ramp (approximately applied load of 70 kN) the composite item suddenly breaks. In particular, the double fixing point breaks the anchoring of both B1 and B2 metallic rings. Figure 7c,d reports some pictures of the broken parts of the item. After the breaking point, the strain values of all sensors recovered to the original values. A not perfect recovery was measured for sensors Y_1_ and Y_2_ probably because they were the closest to the break region. From this consideration, we believe that the anomalous situation occurred during test B at 170 s is due to a partial breaking of the item.

A better comparison between different load tests was conducted in the Figure 9a,b for sensor Y_5_ and for sensors C_1_ and C_4_, respectively. The response of sensors Y_5_ and C_1_ in different test ramps show a very good agreement. With reference to sensor Y_5_, the maximum difference occurs between test B and test C curves at 210 s and is less than 5%. Differently, the sensor C_4_ clearly demonstrated that its behavior during the load test is completely modified by the event during test B, confirming the hypothesis of partial breaking.

In order to demonstrate the capability of FBG sensors technology to be used as real-time tool for structural health monitoring of composite items in aeronautic applications, Figure 10a,b plots 3D real-time maps vs. *y*-axis and *c*-axis, respectively. The map images give a clear understanding of the complexity of the investigated surface in terms of the strain value and of the rate of changes along the spatial axis (*y*- and *c*-axis). Also, the Figure 10b clearly shows the capability to identify the anomalous situation occurs at 170 s and thus shows the capability to act as an early warning monitoring system.

Finally, Figure 11a,b shows 2D plots of the strain profile, versus *y*-axis and *c*-axis, respectively, in different load states during test ramp. This analysis is still focused on the test B section and for three load states, at 90 s, 130 s and 170 s, before the breaking event, and the last one at the end of the loading ramp (after the breaking) at 210 s. Concerning the y-strain profiles, for comparison purposes the Figure 11a also plots the strain profile from numerical results (as reported in Figure 5) on the sensors line and in the same load states. It shows impressive agreement between numerical and experimental results in the center on the composite item. Some differences are evident around the sensors Y_6_ where the experiments return strain values much less that the FEM based expected one. We believe that such behavior is due to the presence of the metallic ring in the fixing point not considered in the numerical model.

The shape of the C-array response in Figure 11b highlights that the region between B1 and B2 exhibits a more critical strain behavior. With increasing applied tension, the dome of the curved region (position of C_1_) is compressed whereas moving to the ends of the arms (to B_1_ and B_2_ fixing points) the positive strain significantly increases. Moreover, the anomaly in the shape after the partial breaking is clearly observable with red curve. In that case, with a load of F = 50 kN, one can expect significant increases of the positive strain of the sensors C_3_ and C_4_: on contrary, the real curve measures lower strain values and an increment of the compression in C_1_.

Based on these results, the proposed FBG monitoring system demonstrated the capability to detect anomalous mechanical behavior as an early warning system.

## 4. Conclusions

In this work, we presented a detailed study devoted to assessing the capability of FBG based optical fiber sensing technology to act as in-situ, remote and real-time monitoring platforms of the load applied to a single component of aircraft landing gear. FBG based optical fiber sensor network involving two FBG arrays of six and four gratings was properly designed, implemented and integrated on a real drag brace. The design and integration of the sensors network were assisted by numerical studies of the composite item selected as target of the present project. Several experimental tests have been done where the composite item was subjected to stress in range 0–70 kN by means of a hydraulic press. Obtained results are in very good agreement with the numerical ones and demonstrate the great potentialities of FBG sensors technology to be employed for remote and real-time load measurements on aircraft landing gears. The proposed FBG monitoring system demonstrated the capability to detect strange behavior as an early warning system and demonstrated the capability to detect breaking of the item when the early warning is ignored.

## Figures and Tables

**Figure 1 sensors-19-02239-f001:**
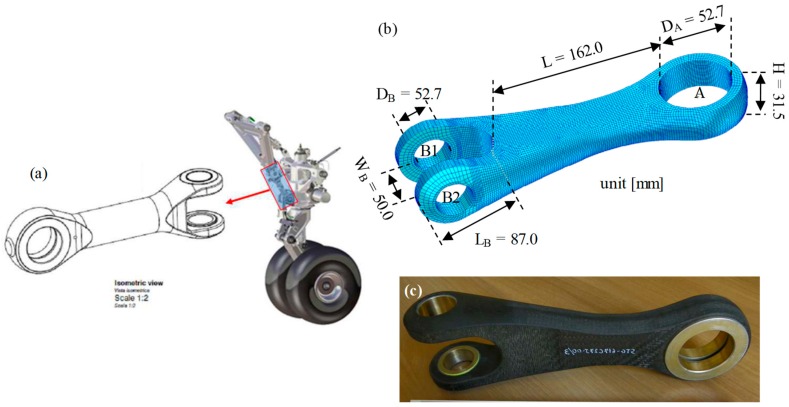
(**a**) Schematic view of the landing gear (LG) and standard drag brace; and (**b**) design and (**c**) picture of the composite drag brace.

**Figure 2 sensors-19-02239-f002:**
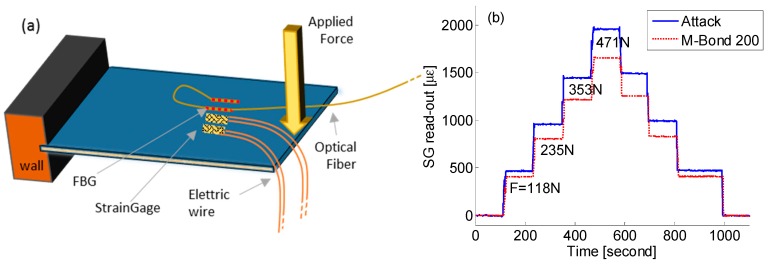
(**a**) Schematic view of the sensorized composite panel test; and (**b**) strain reading of two strain gauges (SGs) fixed by means of Attack and M-Bond 200 adhesive, respectively, for different load states.

**Figure 3 sensors-19-02239-f003:**
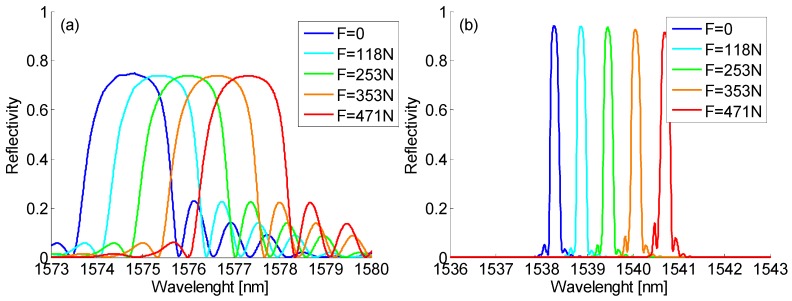
Fiber Bragg Grating (FBG) spectra in different strain states: (**a**) a 1.0 mm long FBG; and (**b**) a 10.0 mm long FBG.

**Figure 4 sensors-19-02239-f004:**
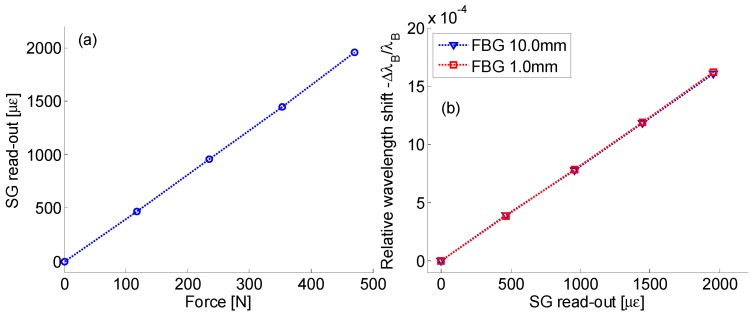
(**a**) Surface strain (from electrical SG) vs. applied force; and (**b**) relative wavelength shifts of FBGs vs. surface strain.

**Figure 5 sensors-19-02239-f005:**
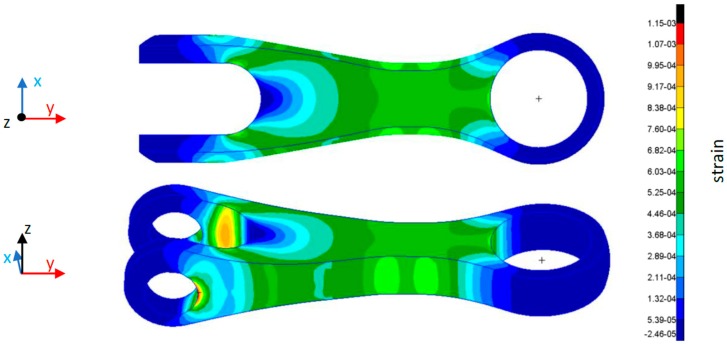
*y*-axis strain by finite element method (FEM) analysis of the composite item.

**Figure 6 sensors-19-02239-f006:**
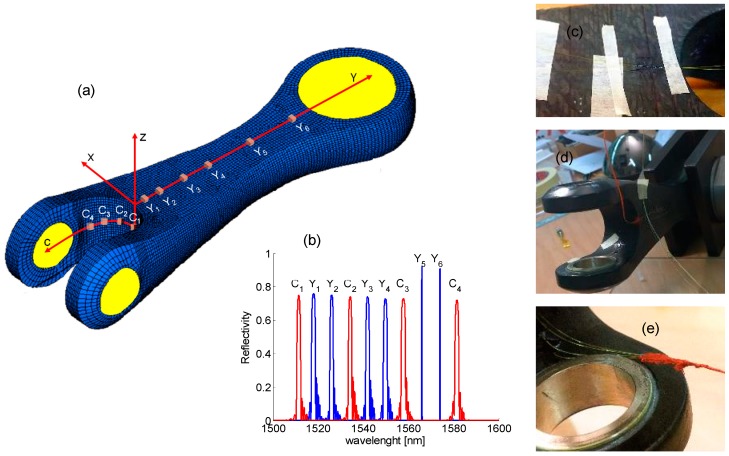
(**a**) Scheme of layout of FBG arrays Y-array and C-array; (**b**) Y-array and C-array spectra; and (**c**–**e**) pictures during the gluing of sensors.

**Figure 7 sensors-19-02239-f007:**
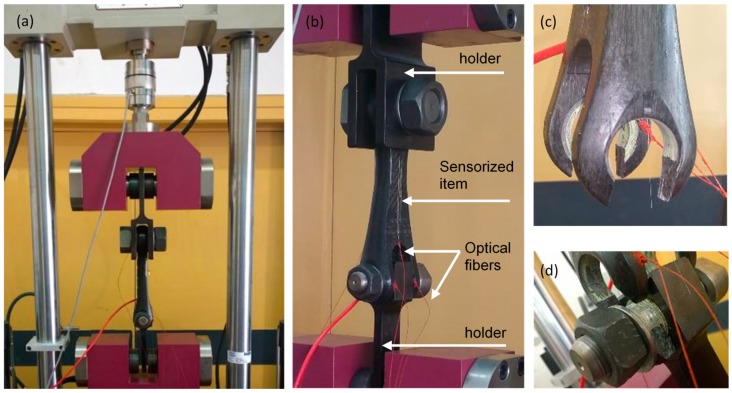
(**a**) and (**b**) Sensorized item fixed in the loading press; (**c**) and (**d**) view of the broken region.

**Figure 8 sensors-19-02239-f008:**
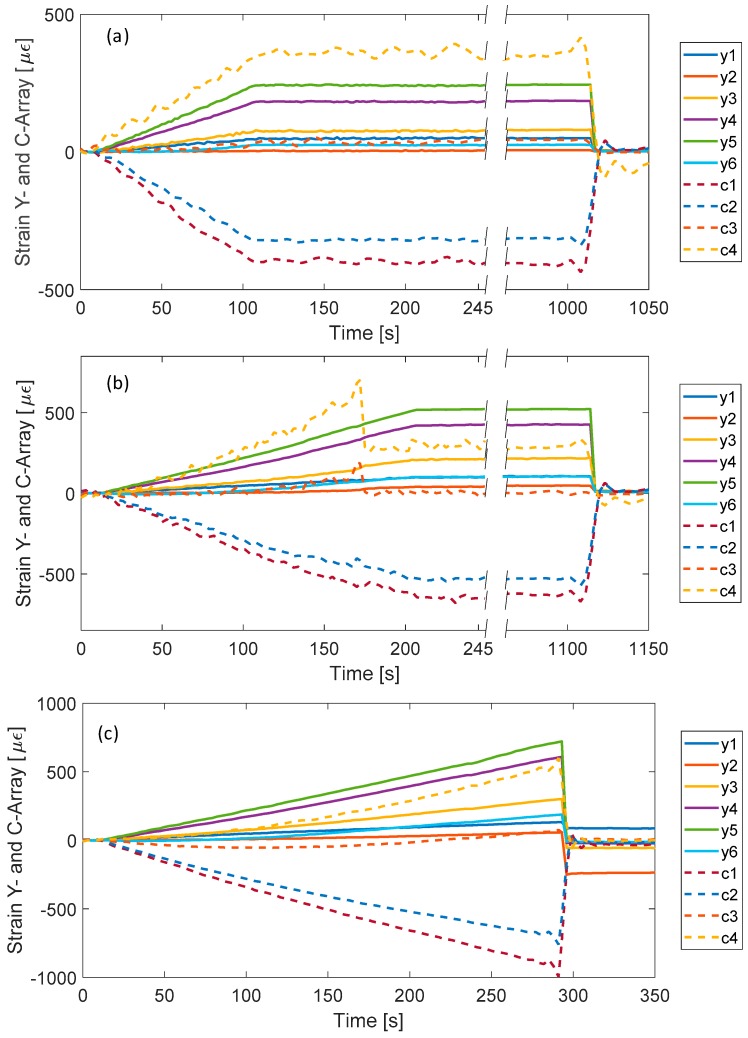
Time response of all FBG sensors in different test sections: (**a**) test A, (**b**) test B, and (**c**) test C.

**Figure 9 sensors-19-02239-f009:**
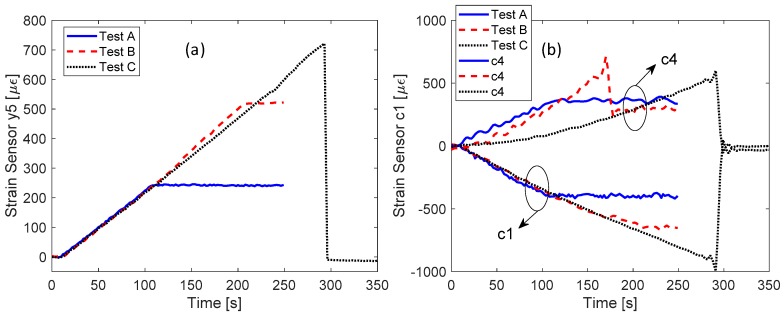
Comparison between the time responses in different test sections for the following sensors: (**a**) sensor Y_5_; and (**b**) sensors C_1_ and C_4_.

**Figure 10 sensors-19-02239-f010:**
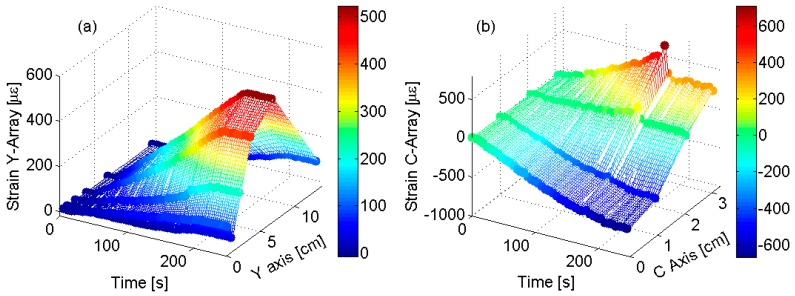
FBG Surface strain profile vs. time: (**a**) strain profile along *y*-axis; and (**b**) strain profile along *c*-axis.

**Figure 11 sensors-19-02239-f011:**
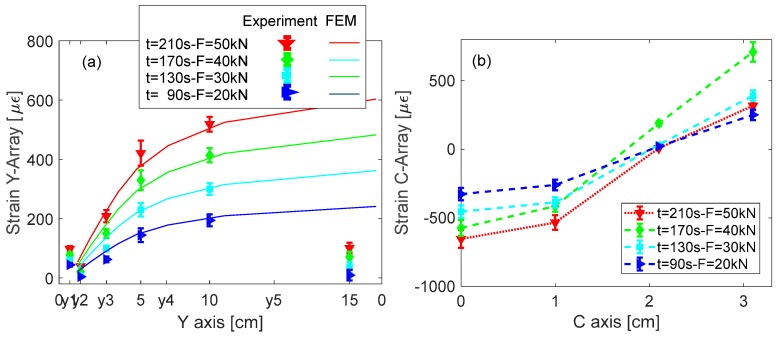
FBG Surface strain profile in different strain states: (**a**) Y-array; and (**b**) C-array.

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
