# Peer review of "Strain Monitoring of a Composite Drag Strut in Aircraft Landing Gear by Fiber Bragg Grating Sensors"

_sensors, 2019, doi:10.3390/s19102239_

Reviewer 1 Report

The article presents FBG sensors for measurement and monitoring of aircraft landing gear made of innovative composite structure. The Authors integrate several fiber optic strain sensors for strain profile characterization. Before the measurement has been carried out, the numerical tests were performed.

On line 87 the Authors claim that “It is worth noting that in most of literature works FBG sensors were applied to large components…”. You can also monitor small components with the Bragg gratings. I suggest referring to the following publications:

Cięszczyk, S., & Kisała, P. (2016). Inverse problem of determining periodic surface profile oscillation defects of steel materials with a fiber Bragg grating sensor. Applied optics, 55(6), 1412-1420.

Jin, B., Zhang, W., Zhang, M., Ren, F., Dai, W., & Wang, Y. (2017). Investigation on characteristic variation of the FBG spectrum with crack propagation in aluminum plate structures. Materials, 10(6), 588.

Zhou, J., Cai, Z., Zhao, P., & Tang, B. (2018). Efficient Sensor Placement Optimization for Shape Deformation Sensing of Antenna Structures with Fiber Bragg Grating Strain Sensors. Sensors, 18(8), 2481.

It would be worth presenting an exemplary stress profile (numerical simulation) on the line with placed sensors.

Line 244, correct the word “emulastion”

Do the authors plan tests during the flight of the aircraft? Is such research possible at all?

The Authors present both simulation and practical realization of the sensor system. Additionally, some tests of the composite module have been performed. In my opinion, the article is interesting and could be publish after a few changes.

Author Response

The authors wish to thank the Reviewers for the useful comments and suggestions. In the following, we provide a point-by-point response to all comments. 

Reviewer:

The article presents FBG sensors for measurement and monitoring of aircraft landing gear made of innovative composite structure. The Authors integrate several fiber optic strain sensors for strain profile characterization. Before the measurement has been carried out, the numerical tests were performed.

On line 87 the Authors claim that “It is worth noting that in most of literature works FBG sensors were applied to large components…”. You can also monitor small components with the Bragg gratings. I suggest referring to the following publications:

Cięszczyk, S., & Kisała, P. (2016). Inverse problem of determining periodic surface profile oscillation defects of steel materials with a fiber Bragg grating sensor. Applied optics, 55(6), 1412-1420.

Jin, B., Zhang, W., Zhang, M., Ren, F., Dai, W., & Wang, Y. (2017). Investigation on characteristic variation of the FBG spectrum with crack propagation in aluminum plate structures. Materials, 10(6), 588.

Zhou, J., Cai, Z., Zhao, P., & Tang, B. (2018). Efficient Sensor Placement Optimization for Shape Deformation Sensing of Antenna Structures with Fiber Bragg Grating Strain Sensors. Sensors, 18(8), 2481.

Authors:

We wish to thank the reviewer for his comment permitting us to improve the paper. Same comments have been added in the introduction and the reference list has been updated accordingly to the reviewer suggestion.

Reviewer:

It would be worth presenting an exemplary stress profile (numerical simulation) on the line with placed sensors.

Authors:

We thank the reviewer for the comment and we are sorry for the unclear description of this issue in the paper. A static stress FEM analysis in provided in section 2.3 where a color map of the surface strain distribution along the main axis of the item when loaded with 50kN is plotted in fig. 5.

Moreover, surface strain numerical profiles (on the line with sensors) are provided in figure 11(a) in different load cases by solid lines, together with experimental results for comparison.

It is worth noting the impressive agreement between numerical and experimental results in the center on the composite item. Some differences are evident around the sensor Y6 where the experiments return strain values much less that the numerical one. We believe that such behavior is due to the presence of metallic ring in the fixing point not considered in the numerical model.  

Note that a more accurate FEM model is possible, however it is time consuming and far from the aim of this work. We modified the paper to better explain the plot of numerical results.

Reviewer:

Line 244, correct the word “emulastion”

Authors:

We thank the reviewer for his comment. The error has been corrected.

Reviewer:

Do the authors plan tests during the flight of the aircraft? Is such research possible at all?

Authors:

The experiments presented in the paper are carried out in laboratory. We would like to repeat the test during the flight of the aircraft ASAP, but it is not yet planed. Test during the flight demand a lot of security requirements that need to be addressed. 

Reviewer:

The Authors present both simulation and practical realization of the sensor system. Additionally, some tests of the composite module have been performed. In my opinion, the article is interesting and could be publish after a few changes.

Authors:

We thank the reviewer for the comment. In the revised version we addressed all suggested changes.

Reviewer 2 Report

The topic described is surely interesting. 

The article is definitively original, however, in the section relating to the description of FBG technology, many parts are found common to well-known reference review-articles.

Nowadays, the technology is well known and is repeatedly described in an ever-increasing number of scientific articles, to be increasingly difficult to propose new schemes to describe the same concepts. However, greater care must be taken to produce paper that leave no doubt.

A text revision is recommended, there are numerous typos that a more careful reading will allow to correct.

In compliance with the SI, it is recommended to separate the number from the unit. In the case of two units of measurement separated by a slash (as it is in case of ratios) the two units must be close to the slash, I suggest eliminating spaces. In the SI, the symbol used for the unit of time is s, which stand for second. The abbreviation sec has no meaning and should not be used in scientific journals.

The paper reports the results of a part of a wider project (CAPRI), as indicated in the acknowledgments. What is the significance of these results when considered in relation to the entire project?

Author Response

The authors wish to thank the Reviewers for the useful comments and suggestions. In the following, we provide a point-by-point response to all comments. 

Reviewer:

The topic described is surely interesting. 

The article is definitively original, however, in the section relating to the description of FBG technology, many parts are found common to well-known reference review-articles.

Nowadays, the technology is well known and is repeatedly described in an ever-increasing number of scientific articles, to be increasingly difficult to propose new schemes to describe the same concepts. However, greater care must be taken to produce paper that leave no doubt.

A text revision is recommended, there are numerous typos that a more careful reading will allow to correct.

Authors:

We thank the reviewer for his comment. We carefully revised the paper and correct typos and other errors.

Reviewer:

In compliance with the SI, it is recommended to separate the number from the unit. In the case of two units of measurement separated by a slash (as it is in case of ratios) the two units must be close to the slash, I suggest eliminating spaces. In the SI, the symbol used for the unit of time is s, which stand for second. The abbreviation sec has no meaning and should not be used in scientific journals.

Authors:

We thank the reviewer for his comment. We carefully revised the paper and correct typos and other errors.

Reviewer:

The paper reports the results of a part of a wider project (CAPRI), as indicated in the acknowledgments. What is the significance of these results when considered in relation to the entire project?

Authors:

The target of the project CAPRI is to develop integrated solutions for an innovative landing gear system for civil aircraft, mainly for the regional transport. The CAPRI project aims to develop innovative technologies for the main components and subsystems of the landing gear of a commercial aircraft, in order to improve the "mission effectiveness" in terms of performance, reliability, maintenance, flight safety, and to develop a strategy for the qualification and certification that makes extensive use of simulation models of elementary parts each duly validated by laboratory tests at full scale or dedicate mock-up.

The WP 04 of the project was devoted to realization of innovative items to reduce the weight of LG, improve the resistance as well as identify appropriate technological platform for SHM of the system.

We believe that the results presented in this paper addressed the aims of the WP 04 of the project.

Reviewer 3 Report

The authors propose and experimentally demonstrate the real-time monitoring of a real drag brace of the aircraft landing gear (LG) using fiber Bragg grating (FBG) series. The strain simulation of drag brace and the measured data are analysed and compared. The paper describes detailed experiments besides designing theory and results thoroughly, so that the paper would be of interest for readers in the field of both LG monitoring and FBG sensing application. This manuscript could be considered for publications in Sensors after the following matters are properly addressed.

1 The drag brace size and dimension should be included in Fig.1 and all other figures.

2 About the temperature compensation in the FBG sensing, the authors referred to their previous paper [7]. But it is not clear how to conduct it, especially for this particular work in the manuscript. It is suggested to make further discussion about this issue.

3 The authors could consider the possible vibration and impact effects for this real drag brace.

4 The manuscript write-up and format is to be improved. It is necessary to carefully edit the manuscript by correcting typos.  For example, Line 360 brage

Author Response

The authors wish to thank the Reviewers for the useful comments and suggestions. In the following, we provide a point-by-point response to all comments.

Reviewer:

The authors propose and experimentally demonstrate the real-time monitoring of a real drag brace of the aircraft landing gear (LG) using fiber Bragg grating (FBG) series. The strain simulation of drag brace and the measured data are analysed and compared. The paper describes detailed experiments besides designing theory and results thoroughly, so that the paper would be of interest for readers in the field of both LG monitoring and FBG sensing application. This manuscript could be considered for publications in Sensors after the following matters are properly addressed.

Authors:

We thank the reviewer for the comment.

Reviewer:

1 The drag brace size and dimension should be included in Fig.1 and all other figures.

Authors:

We thank the reviewer for the comment. The figure 1 is changed with dimension of the item.

Reviewer:

2 About the temperature compensation in the FBG sensing, the authors referred to their previous paper [7]. But it is not clear how to conduct it, especially for this particular work in the manuscript. It is suggested to make further discussion about this issue.

Authors:

We thank the reviewer for the comment. We added some comments about the temperature compensation in section 2.3. To this aim, one more FBG sensor, not fixed to the item, was included in the experimental test and read by the same optoelectronic setup. The last FBG was adopted for the compensation of the thermal changes during the experimentation. To this aim we supposed that the item as well as all the glowed FBGs are in the same thermal state. During the tests temperature changes lower than 3°C were measured.   

Reviewer:

3 The authors could consider the possible vibration and impact effects for this real drag brace.

Authors:

We thank the reviewer for the comment. This is a very interesting topic but currently we did not consider it. We just consider static strain state In future we planned to enable the dynamic measurement of the effect of vibration.

Reviewer:

4 The manuscript write-up and format is to be improved. It is necessary to carefully edit the manuscript by correcting typos.  For example, Line 360 brage

Authors:

We thank the reviewer for his comment. We carefully revised the paper and correct typos and other errors.